# Hodgkin Lymphoma and Hairy Cell Leukemia Arising from Chronic Lymphocytic Leukemia: Case Reports and Literature Review

**DOI:** 10.3390/jcm11164674

**Published:** 2022-08-10

**Authors:** Matteo D’Addona, Valentina Giudice, Luca Pezzullo, Giuseppe Ciancia, Carlo Baldi, Marisa Gorrese, Angela Bertolini, Annapaola Campana, Lucia Fresolone, Paola Manzo, Pio Zeppa, Bianca Serio, Carmine Selleri

**Affiliations:** 1Hematology and Transplant Center, University Hospital “San Giovanni di Dio e Ruggi d’Aragona”, 84131 Salerno, Italy; 2Department of Medicine, Surgery and Dentistry, University of Salerno, 84081 Baronissi, Italy; 3Anatomy Pathology Unit, University Hospital “San Giovanni di Dio e Ruggi d’Aragona”, 84131 Salerno, Italy; 4Department of Clinical Pathology, Insubria University, 21100 Varese, Italy

**Keywords:** Richter’s transformation, chronic lymphocytic leukemia, non-Hodgkin lymphoma, Hodgkin lymphoma

## Abstract

Richter’s syndrome represents the progression of chronic lymphocytic leukemia (CLL) to more aggressive diseases, most frequently diffuse large B-cell lymphoma, while Hodgkin’s lymphoma (HL) and hairy cell leukemia (HCL) are rarely described. The first case involved a 67-year-old man with a diagnosis of a high-risk stage-II CLL treated with rituximab and ibrutinib, developed a HL nodular sclerosis variant after three months of therapy for CLL. After achieving a complete remission for HL and ibrutinib cessation because of drug-related cardiotoxicity, the patient relapsed after five months off-therapy and died due to disease progression after two cycles of brentuximab-vedotin. The second case involved an 83-year-old female with a diagnosis of stage-IV CLL treated with rituximab plus bendamustine who developed a HCL eight years later. Pentostatin was unsuccessfully employed as upfront HCL therapy, and the patient was then switched to rituximab while in remission for CLL. In conclusion, Richter’s transformation risk rate might be higher in patients treated with novel targeted therapies, and multiparametric flow cytometry and lymph node biopsy at relapse could help in early identifying small clones. The treatment of predominant neoplasia is mandatory, and disease-specific drugs are administered; however, clinical efficacy might be lower in these patients.

## 1. Introduction

B-cell chronic lymphocytic leukemia (CLL), a clonal malignant hematological disease of mature B lymphocytes, is the most frequent leukemia in adults and the elderly in Western Countries and is characterized by various biological features and clinical outcomes, mostly with indolent courses, with 5-year overall survivals ranging from 23% to 93% [1,2,3,4,5]. More than half of CLL patients do not require a pharmacological treatment throughout their lifetime, while a third of subjects need to receive standard chemotherapy, and 3–15% of cases evolve into an aggressive disease known as Richter’s syndrome [2,3,4,6,7,8,9]. Diffuse, large B-cell lymphoma (DLBCL) is the most frequent histological type, while classical Hodgkin’s lymphoma (cHL), lymphoblastic lymphoma, hairy cell leukemia (HCL), and high-grade T cell lymphoma are more rarely described [10,11,12,13,14,15,16]. The neoplastic transformation from B-cell CLL to cHL is reported only in 0.5% of all cases, and pathogenesis is still unclear as CLL and cHL might co-exist or one disease might arise from the other [17,18]; however, patients displayed a worse outcome [8]. HCL is a rare chronic lymphoproliferative disorder of mature B lymphocytes, accounting for less than 2% of all leukemias in adults with an indolent course and an 8-year overall survival of 85% [19]. Usually, HCL is associated with the occurrence of a second neoplasia, especially non-Hodgkin lymphomas, while simultaneous HCL and CLL are extremely rare, with only a few cases reported [19,20,21,22,23]. However, no consensus guidelines for treating these secondary lymphomas or leukemias are present. Here, we reported two CLL cases who developed a cHL or an HCL as a second malignancy, and their clinical management and outcomes.

## 2. Case Report 1: CLL and cHL

A 67-year-old man arrived at our observation for absolute lymphocytosis (4850 cells/µL) and lateral cervical lymphadenopathy without any other symptoms in December 2019, and a diagnosis of CLL stage II was made according to 2016 World Health Organization criteria and the RAI staging system [24,25]. A mild thrombocytopenia was also present (141,000 platelets/µL) with normal blood counts (hemoglobin levels, 13.7 g/dL; white blood cells, 9780 cells/µL; and absolute neutrophil count, 4108 cells/µL), normal lactate dehydrogenase levels (348 mU/mL), and normal erythrocyte sedimentation rate (4 mm/h). CLL cells were CD19^+^CD20^+^CD5^+^CD23^+^CD200^+^ by flow cytometry analysis, and he had *TP53* mutation and unmutated immunoglobulin heavy chain (*IGHV*) status, identifying a high-risk disease. CT scan imaging showed multiple bilateral lymphadenopathies in the lateral cervical (maximum diameter, 3.2 cm) area, paratracheal and abdominal region, and splenomegaly (longitudinal diameter, 16.5 cm), with a radiotracer uptake of 4.9 at Positron Emission Tomography (PET) scan. The patient was monitored until July 2020, when a PET/CT scan re-evaluation was performed showing an increase in lymph node and spleen size and in radiotracer uptake. Therefore, therapy with rituximab, an anti-CD20 monoclonal antibody, and ibrutinib, a Bruton’s tyrosine kinase (BTK) inhibitor, was commenced. After three months of treatment, no improvements were observed, and PET/CT scan re-evaluation displayed a further increase in lymph node size and radiotracer uptake (maximum SUV, 11). A lateral cervical lymph node biopsy was performed showing an altered tissue architecture characterized by an inflammatory infiltrate predominantly composed of CD3^+^ T lymphocytes, and rare PAX5^+^CD20^-^CD23^-^ B cells, eosinophils, and plasma cells. Large mono-/bi-nucleated Hodgkin and Reed-Sternberg-like cells were identified, and they were positive for CD30, PAX5, IRF4/MUM1, and BCL6, with increased PD-L1 expression but negative for CD20, CD10, CD3, and CD23 (Figure 1). A histological definition of the cHL nodular sclerosis variant was made, and the patient received a diagnosis of cHL stage-II A/X according to the Ann Arbor system [26]. At the time of progression, the patient had lymphopenia (470 cells/µL) and mild thrombocytopenia (105,000 platelets/µL), with normal blood counts (hemoglobin levels, 13 g/dL; white blood cells, 6050 cells/µL; and absolute neutrophil count, 4901 cells/µL), increased lactate dehydrogenase levels (817 mU/mL), and a high erythrocyte sedimentation rate (25 mm/h).

The R-CHOP regimen (rituximab, cyclophosphamide, doxorubicin, vincristine, and prednisone) was initiated, and ibrutinib administration was continued. After six cycles of therapy, the patient achieved a complete remission (CR), and radiotherapy was performed as consolidation. Ibrutinib was stopped in July 2021, after one year of treatment because of cardiotoxicity requiring hospitalization for heart failure, as reported as BTK inhibitor-related adverse event [27]. After five months off-therapy, PET/CT scan re-evaluation showed a disease relapse and brentuximab-vedotin, and an anti-CD30 monoclonal antibody combined with a potent anti-microtubule agent monomethyl auristatin E was initiated. However, the patient died for disease progression after only two cycles of treatment, with an overall survival of 23.4 months. An autopsy was not performed because of the patient’s family’s decision.

## 3. Case Report 2: CLL and HCL

An 83-year-old female with a history of arterial hypertension, atrial fibrillation, and Hashimoto’s disease diagnosed in 1966 and treated with tapazole for 38 years arrived at our observation in 2014 with a diagnosis of CLL made in another Institution. Flow cytometry immunophenotyping on peripheral blood at diagnosis identified a CLL clone positive for CD5, CD20, CD23, and SmIg κ (κ/λ, 99%/1%), and negative for CD38, CD200, CD10, and CD103. Fluorescence in situ hybridization (FISH) analysis did not detect any chromosomal abnormalities. BM aspirate showed the presence of 29% of small size lymphocytes with compact chromatin by light microscopy and CD5^+^CD20^+^CD23^+^SmIg κ^+^ by flow cytometry. The patient had grade I normocytic anemia (hemoglobin, 11.5 g/dL; mean corpuscular volume, 81.9 fL) and mild thrombocytopenia (platelet count, 122,000/µL) without lymphocytosis (absolute lymphocyte count, ALC, 5710 cells/µL). Spleen enlargement (bipolar diameter, 181 mm) was detected by ultrasound with a radiotracer uptake of 4.7 by PET scan. Moreover, several lymph node enlargements were observed by CT scan: diaphragmatic (12 × 8 mm); hepatic hilum (23 × 12 mm); celiac; pericaval; lumbar aortic; mesenteric; and parasternal (radiotracer uptake, SUV, 1.5). The patient was diagnosis with CLL stage-IV disease, and rituximab and bendamustine were started for a total of six cycles achieving a CR.

After one year of off-therapy, PET/CT scan re-evaluation was performed showing a persistent radiotracer uptake by the enlarged spleen (SUV, 2; diameters, 118 × 73 mm; area, 58 cm^2^); however, no pathological lymph nodes were found, and the patient remained under observation. In September 2019, five years from diagnosis, pathological CD19^+^CD5^+^CD23^+^SmIg κ^+^ lymphocytes were observed by flow cytometry accounting for 18% of total circulating cells. Blood counts were normal, and no signs of lymphadenopathies were present. In February 2022, because of worsening blood counts and increased spleen size (bipolar diameter, 198 mm), a PET/CT scan re-evaluation was carried out that confirmed splenomegaly (bipolar diameter, 200 mm) with increased radiotracer uptake (SUV, 5.2). No pathological lymph nodes were detected. Peripheral blood flow cytometry immunophenotyping displayed a total of 58% lymphocytes, composed as follows: 17% of total CD3^+^ T cells; 6% of CD4^+^ and 10% of CD8^+^ cells with an abnormal ratio of 0.6; 18% of CD7^+^; 17% of CD5^+^; and 70% of CD19^+^ without any CD5^+^CD19^+^ B cells. A more detailed immunophenotyping was carried out on B cells, showing negativity for CD10, CD5, CD23, CD43, and CD38, and positivity for CD20, CD22, FMC7, CD103, CD11c, CD25, CD49d, CD200, and SmIg κ (Figure 2). No chromosomal abnormalities were detected by FISH analysis during disease progression. A diagnosis of HCL was made, and pentostatin was initiated without hematological improvements after eight cycles of chemotherapy, and, at the time of writing, the patient was switched to rituximab while in CR for CLL.

## 4. Discussion

In 1928, Richter’s syndrome has been described for the first time by Maurice N. Richter in a CLL case with lymphadenopathies who evolved with liver and spleen enlargements, resulting in patient’s dead [19]. In most cases, CLL transforms into an aggressive DLBCL; however, prolymphocytic leukemia, HL, lymphoblastic lymphoma, and HCL have also been described [10].

HL arising from CLL is documented in less than 0.4% of all cases, especially in older males presenting with systemic symptoms, lymphadenopathies, and liver and spleen enlargements [12]. CLL prognostic factors, such as advanced stage, *IGHV* mutational status, and cytogenetics abnormalities, are not related to clinical outcomes of patients with HL transformation; conversely, HL prognostic features, including lactate dehydrogenase levels, international prognostic score (IPS), and Richter Scoring System values could identify high-risk CLL patients with HL transformation [28]. Our HL-transformed CLL patient was older than the reported ones (67 years old) (Table 1), and had lymphadenopathies and spleen enlargement at the time of HL transformation without systemic symptoms; IPS was 3, and was an early unfavorable HL disease. Moreover, the patient had two negative CLL prognostic markers, such as *TP53* mutation and unmutated *IGHV* status, identifying a high-risk disease. Therefore, our patient showed unfavorable prognostic markers for both CLL and HL.

Two different post-CLL HL variants can be observed: type I is characterized by few Hodgkin and Reed-Sternberg (HRS) cells within a CLL background, suggesting the neoplastic transformation of a CLL clone into a HRS cell, especially when these latter express B-cell surface markers [12,14,22]. In type II, HRS cells are typically surrounded by an inflammatory infiltrate, while CLL clones are separated from the HL pabulum, supposing an independent origin of the second neoplasia, either treatment-related or a de novo disease [12,20,21]. The most common post-CLL cHL histological variant is mixed cellularity, while nodular sclerosis, lymphocyte-depleted, and lymphocyte-rich forms are less common [7,8,9,10]. Our patient was diagnosed with a nodular sclerosis variant with high expression of PD-L1 on HRS cells, which were surrounded by the typical inflammatory infiltrate mostly constituted by CD3^+^CD5^+^ T cells and fewer CD20^-^CD23^-^PAX5^+^ B cells, eosinophils, and plasma cells.

Different pathogenetic hypotheses have been proposed for Richter’s transformation to HL, including a chronic Epstein–Barr virus infection that might induce a prolonged immunosuppression, thus favoring CLL progression to a high-grade lymphoma [7,10,31]. Immunosuppression could play a pivotal role, as fludarabine and purine analogues cause a lasting T cell reduction and are associated with a shorter overall survival of patients with Richter’s disease [32]. Recently, few cases of Richter’s transformation have been described in CLL patients under ibrutinib treatment, with a median time-to-onset of 15.5 months; however, overall survival rates are similar between patients with Richter’s transformation previously treated with ibrutinib and those who did not receive this small molecule drug. Therefore, there are no conclusive data regarding the risk of Richter’s transformation under BTK therapies. In our patient, Richter’s transformation was observed only after three months under ibrutinib, earlier than that of other reported cases and that of non-ibrutinib treated subjects (3 vs. 15.5 vs. 10 months, respectively); however, overall survival was similar to that of non-ibrutinib treated patients (23 vs. 21 months, respectively), as already reported in the literature [33].

To date, there are no current guidelines for the treatment of HL arising from CLL, even though high-dose chemotherapies are usually employed in fitting patients. The most commonly used chemotherapy regimens are doxorubicin (Adriamycin), bleomycin, vinblastine, and dacarbazine association; cyclophosphamide, vinblastine, procarbazine, and prednisone chemotherapy; or other cytotoxic drugs in combination with rituximab, an anti-CD20 monoclonal antibody [34]. However, the efficacy of those regimens is low, with a reported overall survival of 0.8 years and a progression-free survival of 0.4 years. Other cases have been treated with lomustine and mitoxantrone, and vinblastine, resulting in a 2-year lasting CR [21], or with mechlorethamine plus vincristine, procarbazine, and prednisone, with poor outcomes [35,36]. Our patient received a R-COMP regimen in combination with ibrutinib, resulting in a progression-free survival of 5 months, similar to that reported in previously published cases. Brentuximab-vedotin was used as a salvage therapy, based on efficacy and safety in refractory/relapsed HL treatment; however, no benefits were documented in our HL-transformed CLL patient as he died after only two cycles of therapy.

HCL is an infrequent indolent lymphoproliferative disorder with an 8-year overall survival of 85%, and its coexistence with CLL is even rarer and anecdotally reported [19,20,21,22,23,37]. Hairy cells derive from post-germinal center B lymphocytes, showing positivity for CD19, CD20, CD22, CD11c, and CD200; negativity for CD5, CD23, CD10, and CD27; and an aberrant expression of CD103, CD123, and CD25 [38]. Moreover, neoplastic cells always harbor a characteristic somatic mutation, V600E, on the *BRAF* gene [39] and frequently carry other chromosomal abnormalities, such as trisomy 5, pericentromeric modifications on chromosome 5 and 2, or abnormalities on chromosome 1q42; however, no recurrent chromosomal alteration is observed [40]. Second neoplasia in HCL patients is common, and non-Hodgkin lymphomas are the most represented [41,42,43,44,45,46,47,48]; conversely, HCL arising from CLL is a rare event, while concomitant HCL and CLL is relatively more frequent (Table 2) [19,49].

The clinical significance of these different manifestations remains unclear as patients show similar outcomes [37]. In our case, the HCL clone arose from a CLL five years earlier treated with rituximab plus bendamustine. No international guidelines are present for the treatment of concomitant HCL and CLL, even though a therapeutic strategy is chosen based on the predominant clone at diagnosis [37,38]. In our case, pentostatin was started because the HCL clone was predominant, while minimal residual disease for CLL was negative; however, the patient was switched to rituximab because of poor hematological improvement. Others have associated a purine analogue, such as fludarabine, cladribine, or pentostatin, with rituximab or ibrutinib, while dabrafenib, a BRAF inhibitor, in association with trametinib, a MEK inhibitor, has been rarely employed [19,51,52]. All these therapeutic choices result in a high response rate and a good prognosis, while minimal residual disease can persist, especially the CLL clone [19].

## 5. Conclusions

In conclusion, Richter’s transformation is a rare event and might be determined by different pathogenetic mechanisms, such as TP53 pathway alterations in HL post-CLL or different unknown causes in other variants (e.g., HCL). However, the clinical guard should be always high, especially with patients treated with novel targeted therapies for which Richter’s transformation risk rate is still unclear. Multiparametric flow cytometry and immunohistochemistry on lymph node biopsy at relapse might help with the early identification of small clones; therefore, a complete wide antibody panel should always be performed and not be limited to minimal residual disease monitoring. The treatment of predominant neoplasia is mandatory, and disease-specific drugs are administered, such as ABVD regimen for HL-transformed CLL; however, the clinical efficacy of these drugs might be lower in patients with Richter’s syndrome. Therefore, larger studies or metanalysis are needed to outline clinical and therapeutic guidelines for the treatment of Richter’s disease.

## Figures and Tables

**Figure 1 jcm-11-04674-f001:**
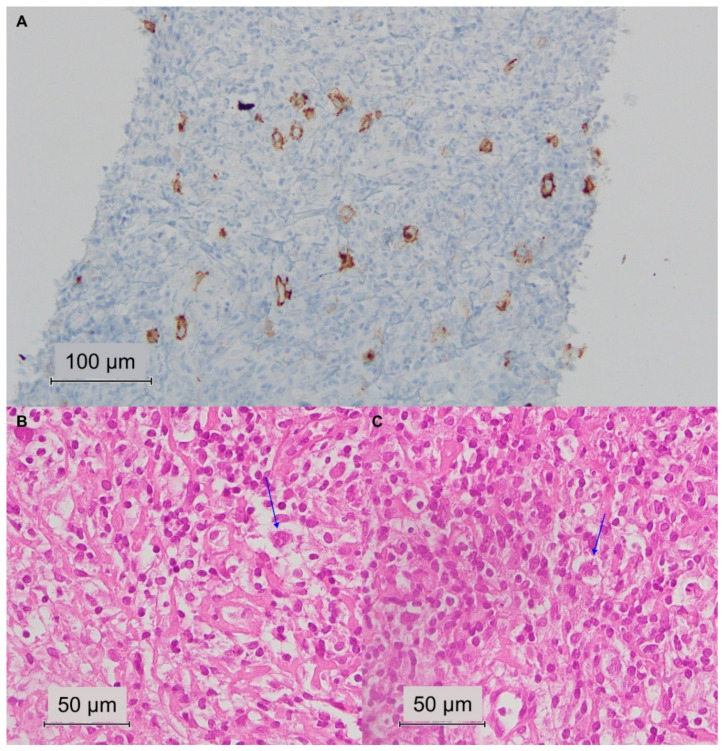
Hodgkin lymphoma arising from chronic lymphocytic leukemia. (**A**) Immunohistochemistry shows CD30^+^ neoplastic cells within the lateral cervical lymph node. (**B**) Hematoxylin and eosin staining displays an altered architecture characterized by an inflammatory infiltrate, and (**B**,**C**) rare large size mono-/bi-nucleated Hodgkin and Reed-Sternberg-like cells (highlighted with blue arrows).

**Figure 2 jcm-11-04674-f002:**
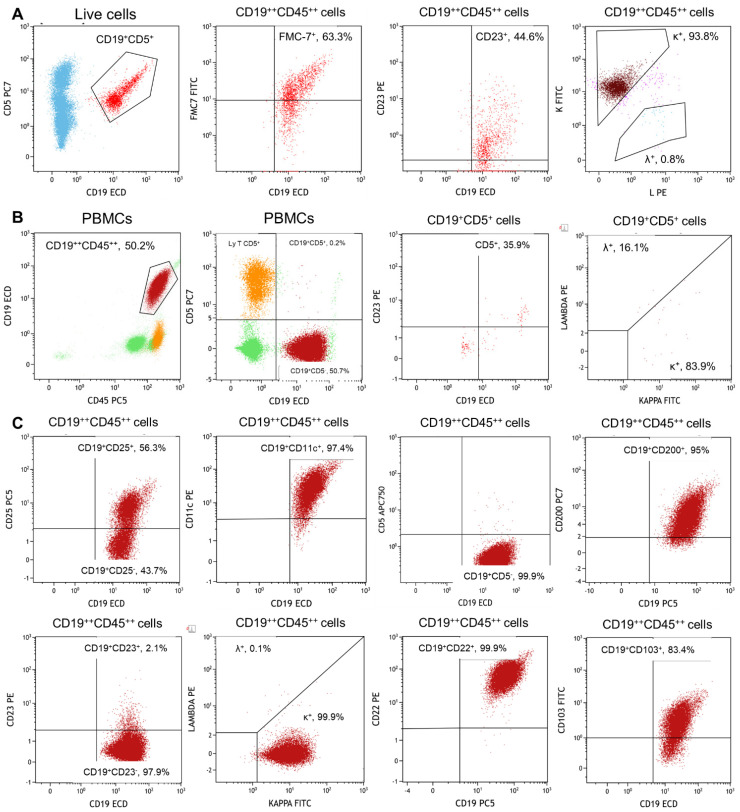
Flow cytometry characterization of chronic lymphocytic leukemia (CLL) and hairy cell leukemia (HCL) clones. (**A**) CLL clone after five years from diagnosis showing positivity for CD19, CD5, CD23, and SmIg κ. (**B**) Minimal residual disease of CLL clone at Richter’s disease diagnosis displaying a small CLL population positive for CD19, CD5, CD23, and SmIg κ. (**C**) HCL clone with negativity for CD10, CD5, CD23, CD43, and CD38, and positivity for CD20, CD22, FMC7, CD103, CD11c, CD25, CD49d, CD200, and SmIg κ.

**Table 1 jcm-11-04674-t001:** Reported cases of HL-transformed CLL.

Reference	N. of Patients	Median Age (Years)	Time-to-Richter’s Transformation (Years)	Survival (Years)
[7]	2	48.4	1	2.4
[8]	1	44	1	0.4
[11]	1	70	0.2	-
[28]	26	67 (45–88)	6.2 (0–24.5)	3.9
[29]	86	65.7 (34–85)	4.3 (0–26)	1.7 (0–14)
[30]	16	58	5.9 (0.8–11.9)	3.3

Abbreviations. HL, Hodgkin lymphoma; CLL, chronic lymphocytic leukemia.

**Table 2 jcm-11-04674-t002:** Reported cases of HCL arising from CLL.

Reference	N. of Patients	Median Age (Years)	Time-to-Richter’s Transformation (Years)	Survival (Years)
[18]	1	75	17	-
[19]	3	52	0	-
[20]	1	74	3	-
[21]	1	72	0	-
[22]	1	83	0	2.8
[50]	5	57.4	0–7	6.7
[23]	6	68.8	0, 3.1, 20	2.9

Abbreviations. HCL, hairy cell leukemia; CLL, chronic lymphocytic leukemia.

## Data Availability

Data are available upon request by the authors.

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
