# Peer review of "Hodgkin Lymphoma and Hairy Cell Leukemia Arising from Chronic Lymphocytic Leukemia: Case Reports and Literature Review"

_jcm, 2022, doi:10.3390/jcm11164674_

Round 1
Reviewer 1 Report
These two case reports present the fate of CLL as incurable disease that eventually transform into a more aggressive disease
Froom the reference it seems that the transformation into HL is much common than it is reported and you highlighted the importance of reassessment every time for other clone and not only the MRD
Though I could be better documented if the genetic profile of the patients as it may give a highlight about the pathogenesis of Richter transformation in those cases
You expanded the description of flow cytometry of the 2 patients and I think , genetic check is important as well.
Lastly , HCL may carry a better prognosis than CLL with excellent 5 years survival would us call it “transformation”??
The attached file shows some grammatic correction’s suggestions and I encourage you for doing language revision.
These 2 cases are very interesting specially in the era of the small molecule treatment of CLL and how we must be cautious in using them
Author Response
These two case reports present the fate of CLL as incurable disease that eventually transform into a more aggressive disease.
Comment 1. From the reference it seems that the transformation into HL is much common than it is reported and you highlighted the importance of reassessment every time for other clone and not only the MRD.
Response to Comment 1. We totally agree with this Reviewer’s comment. Indeed, also HCL arising from CLL is considered extremely rare; however, Giné et al. demonstrated that by retrospectively analyzing HCL cases, it was possible to identify additional composite lymphomas (From Giné E, Bosch F, Villamor N, Rozman M, Colomer D, López-Guillermo A, Campo E, Montserrat E. Simultaneous diagnosis of hairy cell leukemia and chronic lymphocytic leukemia/small lymphocytic lymphoma: a frequent association? Leukemia. 2002 Aug;16(8):1454-9: “Since 1991, all the cases of HCL diagnosed in our center were systematically screened for the presence of additional B cell populations using an extensive panel of monoclonal antibodies. Using this approach, three patients with the simultaneous existence of a HCL and CLL/SLL were identified.” Therefore, it is important to perform a complete immunophenotyping or a lymph node biopsy in suspected relapse rather than looking only at the MRD.
Comment 2. Though I could be better documented if the genetic profile of the patients as it may give a highlight about the pathogenesis of Richter transformation in those cases. You expanded the description of flow cytometry of the 2 patients and I think, genetic check is important as well.
Response to Comment 2. We thank the Reviewer for this point. In the first case, TP53 and IGHV mutational status assessment was performed by next-generation sequencing, showing mutated TP53 and unmutated IGHV. In the second case, fluorescence in situ hybridization (FISH) analysis was carried out at diagnosis and at disease progression as per clinical practice, showing no chromosomal alterations in both cases (no del(17p) found). This information was inserted accordingly in the revised manuscript.
Comment 3. Lastly, HCL may carry a better prognosis than CLL with excellent 5 years survival would us call it “transformation”??
Response to Comment 3. We agree with the Reviewer’s comment, and we have avoided the term “transformation” related to HCL arising from CLL.
Comment 4. The attached file shows some grammatic correction’s suggestions and I encourage you for doing language revision.
Response to Comment 4. We thank the Reviewer for this comment, and we have carefully checked the manuscript for grammatical errors.
Comment 5. These 2 cases are very interesting specially in the era of the small molecule treatment of CLL and how we must be cautious in using them.
Response to Comment 5. We thank the Reviewer for this positive comment on our work.
Reviewer 2 Report
In this manuscript, Matteo D'Addona et al describe 2 interesting CLL cases who later on in the disease course developed HL and HCL. The cases are interesting because such instances are rare and is informative to read on their clinical management.
Few comments:
1. Introduction line 41 maybe better "ranging from 23 to 93%"
2. In the case 2 it would be informative to show the CBC counts at disease onset of the patient as provided for case N.2
3. Line 163 better specify that the Authors are referring to the post-CLL cases while saying :"The most common cHL histological variant is mixed cellularity.."
4. A Table summarizing the previous CLL cases developing HL/HCL reported in the literature and cited in the references would improve the value of this manuscript
Author Response
In this manuscript, Matteo D'Addona et al describe 2 interesting CLL cases who later on in the disease course developed HL and HCL. The cases are interesting because such instances are rare and is informative to read on their clinical management.
Response to General Comment. We thank the Reviewer for this positive feedback.
Few comments:
Comment 1. Introduction line 41 maybe better "ranging from 23 to 93%"
Response to Comment 1. We thank the Reviewer for this suggestion, and we have changed the text accordingly.
Comment 2. In the case 2 it would be informative to show the CBC counts at disease onset of the patient as provided for case N.2.
Response to Comment 2. We thank the Reviewer for this comment, and we have added missing information both at diagnosis and disease progression.
On page 2, lines 62-65, the following text was added “A mild thrombocytopenia was also present (141,000 platelets/µL) with normal blood counts (hemoglobin levels, 13.7 g/dL; white blood cells, 9,780 cells/µL; and absolute neutrophil count, 4,108 cells/µL), normal lactate dehydrogenase levels (348 mU/mL), and normal erythrocyte sedimentation rate (4 mm/h).”
On page 2, lines 84-88, the following text was added “At the time of progression, the patient had lymphopenia (470 cells/µL) and mild thrombocytopenia (105,000 platelets/µL), with normal blood counts (hemoglobin levels, 13 g/dL; white blood cells, 6,050 cells/µL; and absolute neutrophil count, 4,901 cells/µL), increased lactate dehydrogenase levels (817 mU/mL), and high erythrocyte sedimentation rate (25 mm/h).”
Comment 3. Line 163 better specify that the Authors are referring to the post-CLL cases while saying: "The most common cHL histological variant is mixed cellularity."
Response to Comment 3. We thank the Reviewer for this suggestion, and we have changed the text accordingly.
Comment 4. A Table summarizing the previous CLL cases developing HL/HCL reported in the literature and cited in the references would improve the value of this manuscript
Response to Comment 4. We thank the Reviewer for this valuable comment that has markedly improved our manuscript.
Two tables were added for reported cases of HL-transformed CLL (Table 1) or reported cases of HCL-transformed HCL (Table 2).
Table 1. Reported cases of HL-transformed CLL.
|
Reference |
N. of patients |
Median age (years) |
Time-to-Richter’s transformation (years) |
Survival (years) |
|
[7] |
2 |
48.4 |
1 |
2.4 |
|
[8] |
1 |
44 |
1 |
0.4 |
|
[11] |
1 |
70 |
0.2 |
- |
|
[28] |
26 |
67 (45-88) |
6.2 (0-24.5) |
3.9 |
|
[29] |
86 |
65.7 (34-85) |
4.3 (0-26) |
1.7 (0-14) |
|
[30] |
16 |
58 |
5.9 (0.8-11.9) |
3.3 |
Abbreviations. HL, Hodgkin lymphoma; CLL, chronic lymphocytic leukemia.
Table 2. Reported cases of HCL arising from CLL.
|
Reference |
N. of patients |
Median age (years) |
Time-to-Richter’s transformation (years) |
Survival (years) |
|
[18] |
1 |
75 |
17 |
- |
|
[19] |
3 |
52 |
0 |
- |
|
[20] |
1 |
74 |
3 |
- |
|
[21] |
1 |
72 |
0 |
- |
|
[22] |
1 |
83 |
0 |
2.8 |
|
[50] |
5 |
57.4 |
0-7 |
6.7 |
|
[23] |
6 |
68.8 |
0, 3.1, 20 |
2.9 |
Abbreviations. HCL, hairy cell leukemia; CLL, chronic lymphocytic leukemia.